

# Molecular phylogeny of porcelain crabs (Porcellanidae: *Petrolisthes* and allies) from the south eastern Pacific: the genera *Allopetrolisthes* and *Liopetrolisthes* are not natural entities

J. Antonio Baeza

Department of Biological Sciences, Clemson University, Clemson, SC, United States
Smithsonian Marine Station at Fort Pierce, Fort Pierce, FL, United States
Departamento de Biologia Marina, Universidad Catolica del Norte, Coquimbo, IV Region, Chile

## ABSTRACT

Porcelain crabs from the closely related genera *Petrolisthes*, *Liopetrolisthes*, and *Allopetrolisthes* are known for their diversity of lifestyles, habitats, and coloration. The evolutionary relationships among the species belonging to these three genera is not fully resolved. A molecular phylogeny of the group may help to resolve the long-standing taxonomic question about the validity of the genera *Allopetrolisthes* and *Liopetrolisthes*. Using both 'total evidence' and single-marker analyses based on a 362-bp alignment of the 16S rRNA mitochondrial DNA and a 328-bp alignment of the Histone 3 nuclear DNA, the phylogenetic relationships among 11 species from *Petrolisthes* (6 species), *Liopetrolisthes* (2 species), and *Allopetrolisthes* (3 species), all native to the south eastern Pacific, were examined. The analyses supported three pairs of sister species: *L. mitra* + *L. patagonicus*, *P. tuberculatus* + *P. tuberculosus*, and *A. angulosus* + *A. punctatus*. No complete segregation of species, according to genera, was evident from tree topologies. Bayesian-factor analyses revealed strong support for the unconstrained tree instead of an alternative tree in which monophyly of the three genera was forced. Thus, the present molecular phylogeny does not support the separation of the species within this complex into the genera *Petrolisthes*, *Liopetrolisthes*, and *Allopetrolisthes*. Taking into account the above and other recent molecular phylogenetic analyses focused on other representatives from the family Porcellanidae, it is tentatively proposed to eliminate the genera *Liopetrolisthes* and *Allopetrolisthes*, and to transfer their members to the genus *Petrolisthes*.

Corresponding author
J. Antonio Baeza,
baeza.antonio@gmail.com

## INTRODUCTION

Among the Decapoda, crabs from the infraorder Anomura MacLeay, 1838 are renowned for their astounding anatomical, ecological, and behavioral diversity (*McLaughlin et al., 2010*; *Osawa & McLaughlin, 2010*; *Tudge, Asakura & Ahyong, 2012*). During the last decade, various phylogenetic studies have supported monophyly of the Anomura, clarified the position of this clade relative to other decapod lineages, and revealed internal relationships

(*Porter, Perez-Losada & Crandall, 2005*; *Ahyong, Schnabel & Maas, 2009*; *Bracken-Grissom et al., 2013*). Recent studies also have uncovered an evolutionary history much more complex than originally recognized (*Schnabel, Ahyong & Maas, 2011*; *Bracken-Grissom et al., 2013*). Furthermore, some systematic studies, combined with behavioral and ecological observations, have exposed the evolutionary basis for most peculiar behaviors and the conditions favoring them (territoriality and vicious agonistic behaviors in *Allopetrolisthes spinifrons*, living in symbiosis with sea anemones—*Baeza, Thiel & Stotz, 2001*; *Baeza, Stotz & Thiel, 2002*; colonization of hydrothermal vents and unique feeding behavior and associated body parts such as bacterophorian setae in the 'yeti crab' *Kiwa hirsuta*—*Macpherson, Jones & Segonzac, 2006*; *Goffredi et al., 2008*; multiple transitions to crab-like forms from hermit crab ancestors—*Tsang et al., 2011*). Our knowledge of the evolutionary history of anomuran crabs has increased substantially; nevertheless, the internal relationships between many genera and families still remain unknown.

Among anomuran crabs of the superfamily Galatheoidea Samouelle, 1819, one of the most species-rich clades of anomurans, the family Porcellanidae Haworth, 1825, is of particular interest. Crabs from the family demonstrate a considerable diversity of lifestyles, body sizes, habitats, and coloration. More than 280 recognized species (*Osawa & McLaughlin, 2010*; *Osawa & Uyeno, 2013*; *Werding & Hiller, 2015*) inhabit intertidal or shallow subtidal, cold-, warm-temperate, subtropical, and tropical rocky and coral reefs. Some species live in large aggregations, whereas, others remain solitarily within shelters (*Antezana, Fagetti & López, 1965*; *Viviani, 1969*; *Baeza & Stotz, 1995*). Species with cryptic coloration usually dwell under rocks or in crevices, but other, more colorful species inhabit sea anemones in shallow temperate or tropical reefs (*Antezana, Fagetti & López, 1965*; *Baeza, Thiel & Stotz, 2001*). Some colorful species are traded in the marine aquarium industry (e.g., *Porcellana sayana*—*Baeza et al., 2013*). The ecological disparity of crabs from this family has already attracted the attention of systematists (*Werding, Hiller & Misof, 2001*; *Hiller et al., 2006*; *Rodríguez, Hernández & Felder, 2006*; *Miranda, Schubart & Mantelatto, 2014*), evolutionary ecologists (*Baeza & Thiel, 2003*; *Baeza & Asorey, 2012*), and ecophysiologists (*Gebauer, Paschke & Anger, 2010*; and references therein). The same diversity suggests that these crabs are ideal model systems to explore the role of environmental conditions in explaining evolutionary innovations in the marine environment. Phylogenetic studies in the family Porcellanidae are warranted because of the implications for evolutionary ecology, conservation biology, and biodiversity.

In the family Porcellanidae, the genus *Petrolisthes* (*Stimpson, 1858*), was originally established to contain various species of porcelain crabs characterized by, among other traits, a rounded or subquadrate carapace (usually about as broad as long), a triangular or trilobate front often prominent and produced beyond the eyes, a basal segment of the antenna not produced forward to meet the anterior margin of the carapace, either not produced inward, or with a distinct in-ward projection forming a partial suborbital margin, ambulatory legs (pereopods) of moderate length with the propodus bearing movable spinules on the posterior margin and with the dactylus ending in a simple spine, and a telson almost invariably composed of seven plates (*Stimpson, 1858*; *Haig, 1955*). The morphology and taxonomic terminology for the group is shown in Fig. S1.

Later, *Haig (1960)* established three new genera, *Allopetrolisthes*, *Liopetrolisthes*, and *Clastotochus* for a few of the 'most aberrant' species within the genus *Petrolisthes*. The combination of characters setting the genus *Liopetrolisthes*, including the type species *L. mitra*, apart from the closely related *Petrolisthes*, *Allopetrolisthes*, and *Clastotochus*, includes a carapace subovate and slightly longer than broad, a front trilobated and strongly produced beyond the eyes, a basal antennal segment lacking a strong anterior projection in contact with the carapace margin, chelipeds small and flattened in relation to the carapace and with the carpus armed with prominent teeth on the anterior margin, and a telson composed of five plates (also see *Weber, 1991*). In turn, *Allopetrolisthes* differs from species belonging to the remaining closely related genera in exhibiting a combination of the following traits: a carapace rounded and approximately as broad as long, a trilobate front sometimes with two supplementary smaller lobes, a weak anterior projection of the basal antennal segment, which slightly excludes the movable segments from the orbit, a dactylus of the ambulatory legs very short and with posterior movable spinules absent or greatly reduced in size, and a telson composed of five plates (*Haig, 1960*).

*Haig*'s (*1960*) suggestion was followed by scientists throughout the 20th century, and her view has been supported by recent taxonomical studies and the list of porcellanid species from the world (cf. *Osawa & McLaughlin, 2010*). On the other hand, based on molecular characters (i.e., a fragment of the 16S mitochondrial rRNA gene), both *Stillman & Reeb (2001)* and *Rodríguez, Hernández & Felder (2006)* have shown that the genus *Petrolisthes*, as currently recognized, is paraphyletic on the basis of the nested positions of members from the genera *Allopetrolisthes* and *Liopetrolisthes*, among a few others (i.e., *Clastotochus*, *Megalobrachium*, and *Parapetrolisthes*). Similarly, the studies of larval characters also suggest that the genus *Petrolisthes* is paraphyletic and can be subdivided in various natural entities (*Osawa, 1995*; *Wehrtmann et al., 1996*; *Hernández, 1999*; *Fujita, Shokita & Osawa, 2002*; *Hernández & Magán, 2012*). Certainly, additional taxonomic studies are needed to resolve outstanding systematic problems within the family Porcellanidae (*Hiller et al., 2006*).

This study represents a contribution to the phylogeny of crabs from the genus *Petrolisthes* and two of its closely allied genera (i.e., *Allopetrolisthes*, *Liopetrolisthes*) restricted to the temperate south eastern Pacific (Fig. 1). I have focused specifically on addressing the hypothesis of monophyly of the three genera above. It was predicted that a molecular phylogeny of the species included within the three genera should segregate the species into three well-supported monophyletic clades. Based upon the large-subunit, 16S mitochondrial rRNA and the Histone 3 [H3] nuclear DNA, a molecular phylogeny of the species native to the temperate south eastern Pacific is presented in order to examine the hypothesis above.

## MATERIAL AND METHODS
### Taxon sampling, ingroups, and outgroup terminals
A total of 11 species in the genus *Petrolisthes* (6 species) and the related two genera *Allopetrolisthes* (3 species) and *Liopetrolisthes* (2 species), all of them native to the south

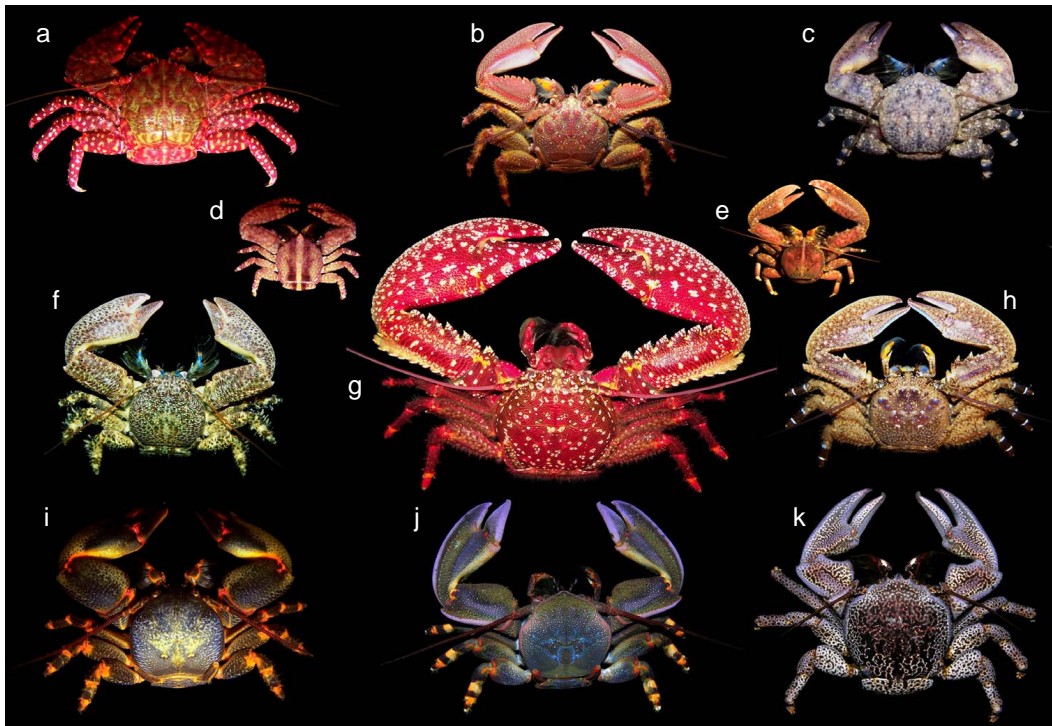

**Figure 1** **Species in the genera *Petrolisthes* (6 species) and the related genera *Allopetrolisthes* (3 species) and *Liopetrolisthes* (2 species), all of them native to the south eastern Pacific.** (A) *Allopetrolisthes spinifrons*, (B) *Petrolisthes tuberculosus*, (C) *Allopetrolisthes angulosus*, (D) *Liopetrolisthes mitra*, (E) *Liopetrolisthes patagonicus*, (F) *Petrolisthes granulosus*, (G) *Petrolisthes desmarestii*, (H) *Petrolisthes tuberculatus*, (I) *Petrolisthes laevigatus*, (J) *Petrolisthes violaceus*, (K) *Allopetrolisthes punctatus*.

eastern Pacific, were included as ingroup terminals in the molecular analyses (Fig. 1). Four other species, *Polyonyx gibbesii*, *Megalobrachium soriatum*, *Pachycheles monilifer*, and *Neopisosoma angustifrons*, were also included in the analyses and used as outgroup terminals. Most crab species were collected by the present author in the coast of Chile. Immediately after collection, specimens were preserved in 95–99% ethanol. The different species were identified using *Haig (1955)*, *Haig (1960)*, *Viviani (1969)*, and *Weber (1991)*. For further details of voucher specimens and GenBank accession information, see Table 1. Altogether, the set of species above was used to reveal the relationship among the genera *Petrolisthes*, *Allopetrolisthes*, and *Liopetrolisthes*.

I also tested for the main hypotheses of monophyly of the genera *Petrolisthes*, *Allopetrolisthes*, and *Liopetrolisthes* (see 'Hypotheses Testing of Monophyletic Clades'). In total, 22 sequences were generated and 11 other sequences were retrieved from GenBank (Table 1).

## DNA extraction, amplification, and sequencing

Total genomic DNA was extracted from pleopods or abdominal muscle tissue using the QIAGEN® DNeasy® Blood and Tissue Kit following the manufacturer's protocol. The polymerase chain reaction (PCR) was used to amplify target regions of one mitochondrial gene (16S (~550 bp)—*Schubart, Neigel & Felder, 2000*) and one nuclear

**Table 1** *Allopetrolisthes—Liopetrolisthes—Petrolisthes* **species and other porcelain crabs used for the phylogeny reconstruction.** The museum catalogue number and the GenBank accession numbers (GenBank) are shown for each species.

| Species | CN | 16S GenBank N | H3 GenBank N |
|---|---|---|---|
| *Allopetrolisthes angulosus* | CU.CC.2016-01-01 | AF260609 | KU641128 |
| *Allopetrolisthes punctatus* | CU.CC.2016-01-06 | AF260615 | KU641133 |
| *Allopetrolisthes spinifrons* | CU.CC.2016-01-07 | AF260617 | KU641134 |
| *Liopetrolisthes mitra* | CU.CC.2016-01-04 | KU641139 | KU641131 |
| *Liopetrolisthes patagonicus* | CU.CC.2016-01-05 | KU641140 | KU641132 |
| *Petrolisthes desmarestii* | CU.CC.2016-01-11 | KU641141 | KU641138 |
| *Petrolisthes granulosus* | CU.CC.2016-01-02 | AF260613 | KU641129 |
| *Petrolisthes laevigatus* | CU.CC.2016-01-03 | AF260606 | KU641130 |
| *Petrolisthes tuberculatus* | CU.CC.2016-01-08 | AF260607 | KU641135 |
| *Petrolisthes tuberculosus* | CU.CC.2016-01-09 | AF260618 | KU641136 |
| *Petrolisthes violaceus* | CU.CC.2016-01-10 | HM352469 | KU641137 |
| *Megalobrachium soriatum* | ULLZ 5262 | DQ865325 | JF900738 |
| *Neopisosoma angustifrons* | ULLZ 5373 | DQ865336 | JF900752 |
| *Pachycheles monilifer* | ULLZ 5348 | DQ865331 | JF900750 |
| *Polyonyx gibbesi* | NA | DQ865341 | JF900736 |

**Notes.**
NA, Not available.

gene (H3 (328 bp)—*Colgan et al., 1998*). For amplification of the 16S and H3 gene segments, I used the primers 16SL2 (5′-TGCCTGTTTATCAAAAACAT-3′) and 16S1472 (5′-AGATAGAAACCAACCTGG-3′) (*Schubart, Neigel & Felder, 2000*) for the 16S gene fragment, and H3AF (5′-ATG GCT CGT ACC AAG CAG ACV GC-3′) and H3AR (5′-ATA TCC TTR GGC ATR ATR GTG AC-3′) for the H3 gene fragment (*Colgan et al., 1998*), respectively.

Standard PCR 25-μl reactions (17.5 μl of GoTaq® Green Master Mix (Promega, Madison, WI, USA), 2.5 μl each of the two primers (10 mM), and 2.5 μl DNA template) were performed on a Peltier Thermal Cycler (DYAD, Norcross, GA, USA) and C1000 Touch™ Thermal Cycler (BIORAD, Hercules, CA, USA) under the following conditions: initial denaturation at 95 °C for 5 min followed by 40 cycles of 95 °C for 1 min, 52–57 °C (depending on the species) for 1 min, and 72 °C for 1 min, followed by chain extension at 72 °C for 10 min. PCR products were purified with ExoSapIT (a mixture of exonuclease and shrimp alkaline phosphatase, Amersham Pharmacia) and then sent for sequencing with the ABI Big Dye Terminator Mix (Applied Biosystems, Foster City, CA, USA) to the Laboratory of Analytical Biology of the National Museum of Natural History, Smithsonian Institution (LAB–NMNH, Maryland) and to the Clemson University Genomics Institute (CUGI–Clemson University, Clemson. South Carolina), which are equipped with ABI Prism 3730xl Genetic Analyzers (Applied Biosystems automated sequencer). All sequences were confirmed by sequencing both strands and a consensus sequence for the two strands was obtained using the software Sequencer 5.4.1 (Gene Codes Corp., Ann Arbor, MI, USA).

**Table 2  Molecular markers including informative sites and maximum likelihood (ML) models selected through AICc criterion as implemented in jModelTest2.** Base frequencies, rate matrix, and gamma shape parameters resulting from jModelTest2 are shown.

| | Gene fragment | |
|---|---|---|
| | **H3** | **16S** |
| **Total sites** | 328 | 362 |
| Informative sites | 50 | 83 |
| **Model** | GTR+G | TVM+G |
| **Base frequency** | | |
| %A | 0.2234 | 0.3507 |
| %C | 0.3116 | 0.1255 |
| %G | 0.2684 | 0.1903 |
| %T | 0.1966 | 0.3336 |
| **Rate matrix** | | |
| [A-C] | 1.3411 | 0.5967 |
| [A-G] | 4.4940 | 8.6307 |
| [A-T] | 4.1603 | 3.0555 |
| [C-G] | 0.6098 | 0.0001 |
| [C-T] | 10.6755 | 8.6307 |
| [G-T] | 1.0000 | 1.000 |
| **Shape parameter** | 0.1780 | 0.2770 |

## Sequence alignment and phylogenetic analyses

Alignment of each set of sequences was conducted using Multiple Sequence Comparison by Log-Expectation in MUSCLE (*Edgar, 2004*) as implemented in MEGA6 (*Tamura et al., 2011*). The alignment of the H3 gene fragment had no indels and was unambiguous. In contrast, the aligned sequences of the 16S gene fragment did contain several indel 'islands'. Therefore, positions that were highly divergent and poorly aligned in the 16S gene segment were identified using the default settings in the software GBlocks v0.91b (*Castresana, 2000*), and omitted from the analyses. After highly divergent positions were pruned, the 16S dataset consisted of 362 bp.

The two datasets were first analyzed with the software jModelTest 2 (*Darriba et al., 2012*), which compares different models of DNA substitution in a hierarchical hypothesis–testing framework to select a base substitution model that best fits the data. For the two gene fragments, the optimal models found by jModelTest 2 (selected with the corrected Akaike Information Criterion [$AIC_c$]) are shown in Table 2. These models were implemented in MrBayes (*Huelsenbeck & Ronquist, 2001*) for Bayesian Inference (BI) analysis and GARLI version 2.1 (available at http://www.molecularevolution.org/software/phylogenetics/garli—*Bazinet, Zwickl & Cummings, 2014*) for maximum likelihood (ML) analysis.

A 'total evidence' analysis (*Grant & Kluge, 2003*) was conducted and thus the two different alignments were concatenated into a single dataset consisting of 15 sequences and 690 bp. However, the dataset was partitioned into two different segments, each with a different model of evolution. Missing data were designated as a '?' in the alignment. All the parameters used for the ML analysis were those of the default option in GARLI. For BI,

unique random starting trees were used in the Metropolis–coupled Markov Monte Carlo Chain (MCMC) (see *Huelsenbeck & Ronquist, 2001*; *Ronquist et al., 2012*). The analysis was performed for 6,000,000 generations. Visual analysis of log-likelihood scores against generation time indicated that the log-likelihood values reached a stable equilibrium before the 100,000th generation. Thus, a burn-in of 1,000 samples was conducted, every 100th tree was sampled from the MCMC analysis obtaining a total of 60,000 trees and a consensus tree with the 50% majority rule was calculated for the last 59,900 sampled trees. The robustness of the ML tree topology was assessed by bootstrap reiterations of the observed data 2,000 times. Support for nodes in the BI tree topology was obtained by posterior probability.

Total evidence analyses enhances the detection of real phylogenetic groups if there is no or minimal heterogeneity among different (e.g., H3 and 16S) datasets (*De Queiroz, Donohue & Kim, 1995*). Therefore, I also conducted separate ML and BI phylogenetic analyses for each gene fragment to reveal any possible discordance in the relationships among the studied species. These phylogenetic analyses using only one gene fragment at a time demonstrated minimal heterogeneity (see 'Results'). Thus, the 'total evidence' analysis has the ability to more accurately reflect phylogenetic relationships in this study (see *De Queiroz, Donohue & Kim, 1995*). Total evidence analyses have been used before to infer the phylogeny of many other clades of marine and terrestrial vertebrates and invertebrates, including marine decapods, e.g., in shrimps (*Duffy, Morrison & Ríos, 2000*; *Anker & Baeza, 2012*; *Baeza, 2013*) and brachyuran crabs (*Hultgren & Stachowicz, 2009*), among others.

### Hypotheses testing of monophyletic clades

I tested if the different species of the genera *Petrolisthes*, *Liopetrolisthes* and *Allopetrolisthes* segregated and formed different genus-specific monophyletic clades. For this purpose, a constrained tree (in which the monophyly of all three genera was enforced) was obtained in MrBayes with the command *constraint*. MCMC searches were run and the harmonic mean of the tree-likelihood value was obtained by sampling the post burn-in, posterior distribution as above. Next, Bayes factors were used to evaluate whether or not there was evidence against monophyly (constrained versus unconstrained trees) according to the criteria of *Kass & Raftery (1995)*. Bayes factors compare the total harmonic mean of the marginal likelihood of unconstrained vs. monophyly-constrained models. A higher value of the Bayes factor statistic implies stronger support against the monophyly of a particular group (*Kass & Raftery, 1995*). Specifically, a value for the test statistic $2 \log_e(B_{10})$ between 0 and 2 indicates no evidence against $H_0$; values from 2 to 6 indicate positive evidence against $H_0$; values from 6 to 10 indicate strong evidence against $H_0$; and values >10 indicate very strong evidence against $H_0$ (*Kass & Raftery, 1995*; *Nylander et al., 2004*).

## RESULTS

The final molecular data matrix was comprised of a total of 690 characters, of which 133 of them were parsimony informative, for a total of 11 ingroup species from the south eastern Pacific pertaining to the genera *Petrolisthes*, *Allopetrolisthes*, and *Liopetrolisthes* and 4 outgroup terminals. Both 'total evidence' molecular phylogenetic trees obtained with different inference methods (ML and BI) resulted in the same general topology (Fig. 2).

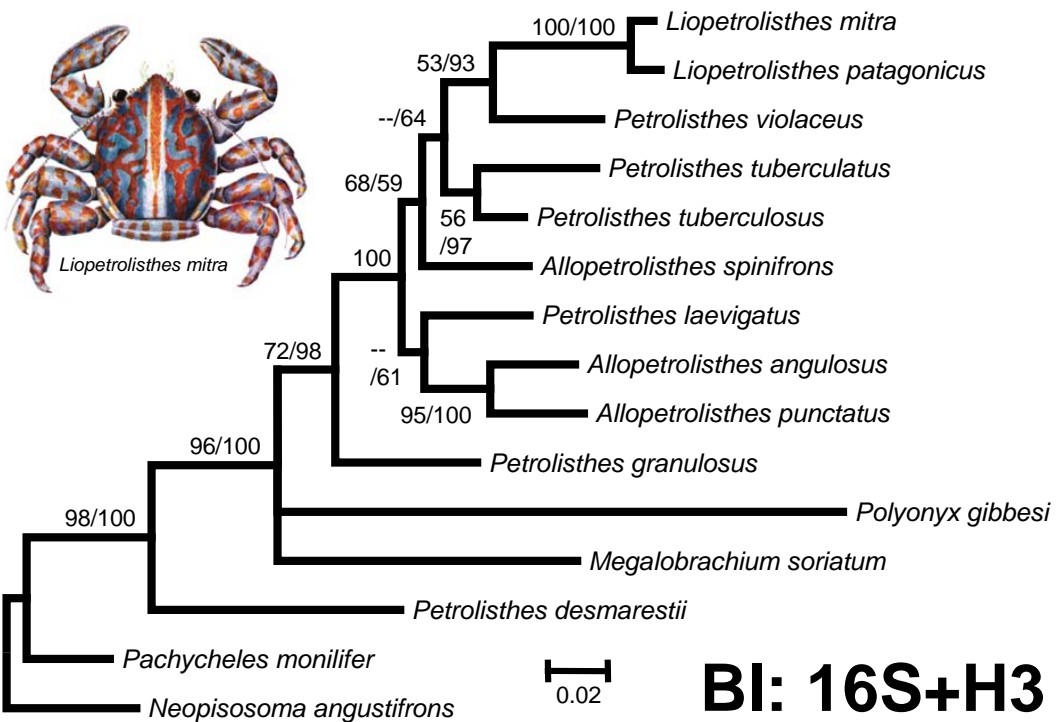

**Figure 2** **'Total evidence' phylogenetic tree obtained from BI analysis of the partial mitochondrial 16S rRNA and nuclear Histone 3 genes for crabs from the *Petrolisthes* and allies.** Numbers above and/or below the branches represent the posterior probabilities from the BI analysis in MrBayes and bootstrap values obtained from ML in GARLI (ML/BI). The general topology of the trees obtained from MP and ML analyses was the same. The inset shows a juvenile of *Liopetrolisthes mitra* after *Meredith (1939)*.

In the two 'total evidence' phylogenetic analyses, with the exception of *P. desmarestii*, species from the genera *Petrolisthes, Allopetrolisthes*, and *Liopetrolisthes* clustered together into a single monophyletic clade strongly supported by a high posterior probability obtained from the BI analysis and was well supported by the bootstrap support values from the ML analysis. Within this clade, *P. granulosus* was revealed as sister to all other species of *Petrolisthes* (excluding *P. desmarestii*), *Allopetrolisthes* and *Liopetrolisthes* from the south eastern Pacific. The status of *A. punctatus* and *A. angulosus* as a pair of sister species is well supported by the BI and ML analyses. The tree topology recovered *P. laevigatus* as sister to *A. punctatus* and *A. angulosus*. Nonetheless, the sister relationship above was poorly supported by a low posterior probability obtained from the BI analysis and bootstrap support values from the ML analysis, respectively. Interestingly, *Allopetrolisthes spinifrons* did not cluster together with the two other congeneric species and its position was not well resolved in the two phylogenetic trees.

In the two phylogenetic analyses, two species from the genus *Liopetrolisthes*, *L. mitra* and *L. patagonicus*, were recovered as well supported sister species. *Petrolisthes violaceus* was recovered as sister to the genus *Liopetrolisthes* with moderate to high support. Lastly, *P. tuberculatus* and *P. tuberculosus* were recovered as sister species with strong support from both ML and BI analyses.

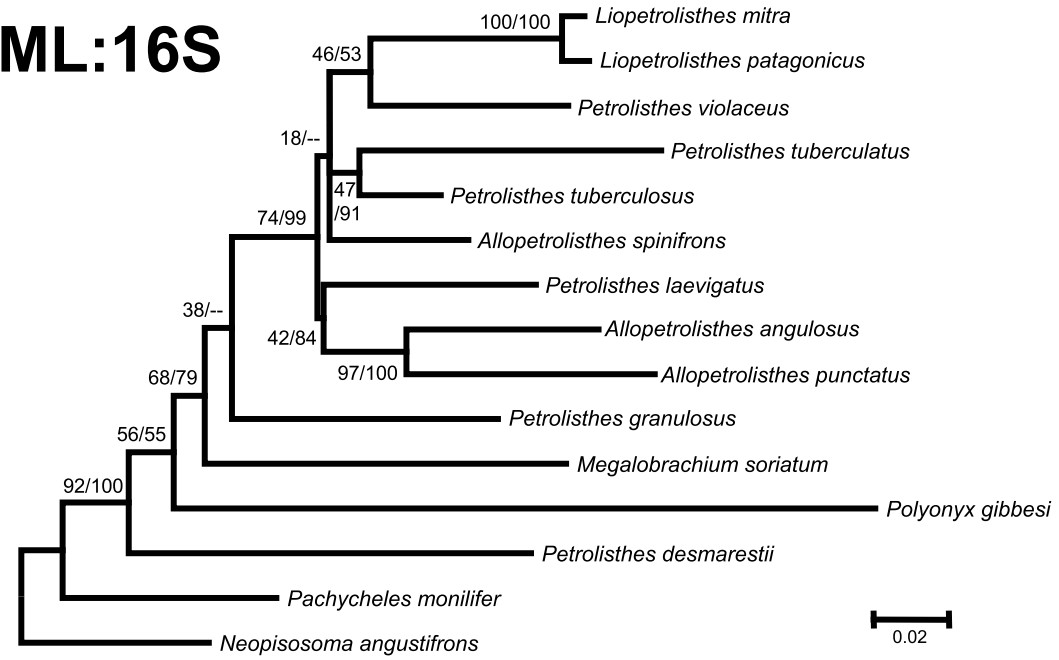

**Figure 3  Phylogenetic tree obtained from ML analysis of the partial nuclear Histone 3 gene for crabs from the *Petrolisthes* species complex, and other selected taxa from the family Porcellanidae.** Numbers above and/or below the branches represent the posterior probabilities from the BI analysis in MrBayes and bootstrap values obtained from ML in GARLI (ML/BI). The general topology of the trees obtained from MP and ML analyses was the same.

Unexpectedly, *P. desmarestii* did not cluster together with other congeneric species. Indeed, *P. desmarestii* was recovered as sister to a clade including all the remaining species of *Petrolisthes, Allopetrolisthes*, and *Liopetrolisthes*, and also containing *Polyonyx gibbesi* and *Megalobrachium soriatum*.

Overall, the 'total evidence' phylogenetic analyses demonstrated that species from the genera *Petrolisthes, Allopetrolisthes*, and *Liopetrolisthes* altogether did not segregate according to genera and did not form well-supported, monophyletic clades, as should be expected according to adult morphology. Similarly, the Bayes factor analysis revealed no support for the separation of the studied species into three different genera (*Petrolisthes, Allopetrolisthes*, and *Liopetrolisthes*). Comparisons of the unconstrained tree (harmonic mean $= -3496.7$) *versus* the tree wherein *Petrolisthes, Allopetrolisthes*, and *Liopetrolisthes* were imposed as monophyletic clades (harmonic mean $= -3402.17$), indicated strong support for the unconstrained tree ($2\ln(B_{10}) = 9.09$).

Phylogenetic trees obtained with ML and BI using only a single, either mitochondrial (16S) or nuclear (H3), marker resulted in similar general topologies (Figs. 3 and 4). As expected, these single-marker phylogenetic trees were less resolved than those produced by the 'total evidence' ML and BI phylogenetic analyses. Nonetheless, the single-gene analyses retrieved various monophyletic clades observed in the 'total evidence' analyses described above. For instance, in both the ML and BI trees based on the 16S and H3 gene fragments, both *L. mitra* and *L. patagonicus*, and *A. angulosus* and *A. punctatus*, were well supported as sister species. *Petrolisthes desmarestii* was recovered as sister to a clade including all

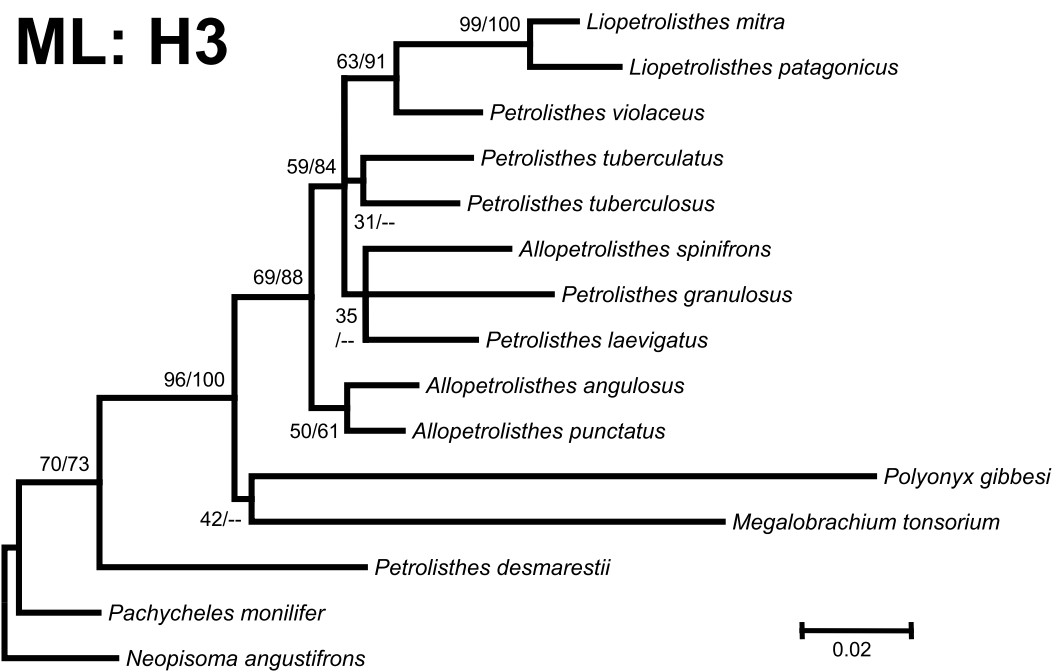

**Figure 4** Phylogenetic tree obtained from ML analysis of the mitochondrial 16S rRNA and nuclear Histone 3 genes for crabs from the *Petrolisthes* and other selected taxa from the family Porcellanidae. Numbers above and/or below the branches represent the posterior probabilities from the BI analysis in MrBayes and bootstrap values obtained from ML in GARLI (ML/BI). The general topology of the trees obtained from MP and ML analyses was the same.

the remaining species of *Petrolisthes*, *Allopetrolisthes*, and *Liopetrolisthes*, and additionally containing *Polyonyx gibbesi* and *Megalobrachium soriatum* (Figs. 3 and 4).

## DISCUSSION

This study presents a two-locus molecular phylogeny of crabs from the genera *Petrolisthes, Liopetrolisthes,* and *Allopetrolisthes*. This pool of species represents the totality of the members from the genus *Petrolisthes* and allies (*Allopetrolisthes* and *Liopetrolisthes*) native to the south eastern Pacific (*Haig, 1960*; *Viviani, 1969*; *Weber, 1991*). The analyses with two different phylogenetic reconstruction methods recognized only one monophyletic group consisting of two species of *Liopetrolisthes* (*L. mitra* and *L. patagonicus*) and also supported two members from the genus *Allopetrolisthes* as sister species (*A. angulosus* and *A. punctatus*). The position of *A. spinifrons*, the remaining congeneric species, was not well resolved. In disagreement with hypotheses based solely upon adult morphology (e.g., *Haig, 1960*), a well-resolved grouping of all of the species belonging to a particular genus and segregation of species from different genera was not revealed by these analyses. Also, Bayesian factors analyses strongly supported unconstrained trees over trees in which monophyly of *Petrolisthes, Liopetrolisthes*, and *Allopetrolisthes* was imposed. Overall, the present results do not support the separation of these species into three different genera as proposed by *Haig (1960)* that was based upon adult morphology alone. Instead, the present study agrees with previous larval and molecular phylogenetic studies

indicating that the division of *Petrolisthes, Liopetrolisthes*, and *Allopetrolisthes* within the Porcellanidae is not natural (*Osawa, 1995*; *Wehrtmann et al., 1996*; *Hernández, 1999*; *Fujita, Shokita & Osawa, 2002*; *Stillman & Reeb, 2001*; *Rodríguez, Hernández & Felder, 2006*; *Hernández & Magán, 2012*). This study argues in favor of future phylogenetic studies using various types of evidence (molecular, adult morphology, larval anatomy) to improve our knowledge of the natural relationships within these species/genera complexes and their position in the Porcellanidae.

The set of species considered in the present study allows a few relevant systematic questions for the group to be addressed. For instance, the two species from the genus *Liopetrolisthes, L. mitra* and *L. patagonicus*, clustered together and formed a well supported monophyletic group. *Liopetrolisthes mitra* inhabits the body surface of the black sea urchin *Tetrapygus niger* from Ancon, Peru (∼11.8°S latitude) to Bahia San Vicente, Chile (∼36°S latitude) while *L. patagonicus* dwells among the spines of the red sea urchin *Loxechinus albus* from Ancon, Peru (∼11.8°S latitude) to the strait of Magellan, Chile (∼54°S latitude) (*Weber, 1991*). This suggests that the genus *Liopetrolisthes* likely diversified in the south eastern Pacific in sympatry although it remains to be addressed whether or not speciation in this genus was host-driven. Importantly, although the genus *Liopetrolisthes* represents a natural clade in the present phylogenetic analyses, its generic status, different from *Petrolisthes*, is not supported as the two species in the genus clustered within a clade that included other members from the genus *Petrolisthes* (also, see below).

The present study also retrieved *P. tuberculosus* and *P. tuberculatus* as a single well supported monophyletic clade. The close relationship between the two species was recognized early on by *Ortmann (1897)* who named them as belonging to the 'Gruppe des *Petrolisthes tuberculatus*,' a view supported by *Haig (1960)*. The two species are characterized by a strongly trilobate front, two narrow lobes that project strongly from the anterior margin of the basal segment of the antennule, and a row of uneven, serrate teeth on the anterior margin of the cheliped carpus (*Haig, 1960*; *Viviani, 1969*). Given the particular distinctiveness of the two species, *Haig (1960)* suggested that they should form a separate genus or subgenus. However, at present, it seems inadvisable to split *P. tuberculosus* and *P. tuberculatus* from the remaining species in the genus until *Petrolisthes* is analyzed from locations worldwide.

Lastly and unexpectedly, *P. desmarestii*, the largest known species of porcelain crab (*Haig, 1960*; *Antezana, Fagetti & López, 1965*), did not cluster together with the other congeneric representatives included in this study. *Petrolisthes desmarestii* was recovered as sister to a clade that included *Petrolisthes, Allopetrolisthes, Liopetrolisthes* as well as *Polyonyx gibbesi* and *Megalobrachium soriatum*. The clustering of *P. desmarestii* with members from the genera *Polyonyx* and *Megalobrachium* likely resulted from incomplete taxon sampling in Porcellanidae. Nonetheless, *Petrolisthes desmarestii* is unique among other congeneric representatives from the south eastern Pacific because of the carapace, covered with fine plications, the presence of a single epibranchial spine on the carapace, a triangular front, a carpus of the cheliped with four or five broad, serrate-edged teeth on the anterior margin, and a manus covered with small flattened tubercles (*Haig, 1960*). The traits above, in particular, the presence of an epibranchial spine on the carapace, teeth on the

anterior margin of the cheliped carpus, and the postero-distal spines on the merus of the first ambulatory pereopod, suggest that *P. desmarestii* belongs to either the 'Gruppe des *Petrolisthes galathinus*' or 'Gruppe des *Petrolisthes lamarcki*' recognized by *Ortmann (1897)*, both containing more than twenty species in the eastern Pacific and western Atlantic (*Haig, 1960*; *Hiller et al., 2006*). A preliminary molecular phylogenetic analysis based on the 16S mtDNA ribosomal gene (Bayesian inference, 95 terminals, GTR+G model, not shown here) provides support for the close relationship between *P. desmarestii* and members from the *P. galathinus* species complex. These results suggest more than a single colonization event in the south eastern Pacific during the evolutionary history of porcelain crabs.

In general, this study has shown that the separation of *Petrolisthes* + *Allopetrolisthes* + *Liopetrolisthes* into three taxonomic entities is not natural based on molecular characters of the studied species set. Crabs from these three genera demonstrate a considerable diversity of lifestyles, body sizes, microhabitats, and coloration (*Haig, 1960*; *Antezana, Fagetti & López, 1965*; *Baeza & Thiel, 2003*; *Baeza & Thiel, 2007*). Studies describing the life history and ecology of *Petrolisthes, Allopetrolisthes*, and *Liopetrolisthes* within a phylogenetic framework are underway (e.g., *Baeza & Thiel, 2003*; *Baeza & Asorey, 2012*; *Gebauer, Paschke & Anger, 2010*). This approach is expected to prove most useful in understanding the role of environmental conditions in driving the evolution of morphological, ecological, and behavioral traits in the marine environment (e.g., *Baeza & Thiel, 2003*; *Baeza & Asorey, 2012*). The present study included only porcellanid species from the south eastern Pacific; nevertheless, the amphi-American nature of *Petrolisthes* and allies (see *Haig, 1960*) suggests that this group might also be a model to study speciation mechanisms, as in other transisthmian clades of fish (*Bermingham, McCafferty & Martin, 1997*), sea urchins (*Lessios, 2008*), caridean shrimps (*Williams et al., 2001*), and brachyuran crabs (*Windsor & Felder, 2014*).

## Proposal of a phylogenetic rearrangement

Taking into account the discussion above and recent molecular phylogenetic analyses focused on other representatives from the family Porcellanidae (i.e., *Weber, 1991*; *Rodríguez, Hernández & Felder, 2006*; *Stillman & Reeb, 2001*; *Hiller et al., 2006*), the following taxonomic rearrangement is tentatively proposed for the south eastern Pacific species hitherto belonging to the genera *Allopetrolisthes* and *Liopetrolisthes*.

Family Porcellanidae Haworth, 1825

*Petrolisthes* Stimpson, 1858

*Petrolisthes angulosus* (Guérin, 1835)

*Petrolisthes punctatus* (Guérin, 1835)

*Petrolisthes spinifrons* (H. Milne Edwards, 1837)

*Petrolisthes mitra* (Dana 1852)

*Petrolisthes patagonicus* (Cunningham, 1871)

## ACKNOWLEDGEMENTS

The author thanks Mr. Helmo Perez for his help with collection of porcelain crabs from the turbid and harsh waters of Chile. Lunden Simpson critically reviewed the English as well as the content and provided helpful comments. Special thanks to Martin Thiel (UCN, Chile) for inviting the author to co-instruct a short course on the biology of caridean shrimps in Coquimbo during 2007 that permitted sampling of specimens. This is contribution number 1026 of the SMSFP. Special thanks to Arthur Anker for permission to use some of his astonishing photographs.

### Funding

The author received no funding for this work.

### Competing Interests

The author declares there are no competing interests.

### Author Contributions

- J. Antonio Baeza conceived and designed the experiments, performed the experiments, analyzed the data, contributed reagents/materials/analysis tools, wrote the paper, prepared figures and/or tables, reviewed drafts of the paper.

### Data Availability

GenBank accession numbers provided in the manuscript in Table 1.

### Supplemental Information

Supplemental information for this article can be found online at http://dx.doi.org/10.7717/peerj.1805#supplemental-information.

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
