# Peer review of "Molecular phylogeny of porcelain crabs (Porcellanidae: Petrolisthes and allies) from the south eastern Pacific: the genera Allopetrolisthes and Liopetrolisthes are not natural entities"

_PeerJ, doi:10.7717/peerj.1805_

## Round 0.1 · original submission · Major Revisions

I have received two generally positive reviews on this paper. However, both reviewers do make some suggestions for improvement of the paper. I suggest you follow their suggestions if possible, or if not provide reasons why this is not your choice.
One exception is the comments by reviewer 2 on the appropriateness of this manuscript to PeerJ. I have reviewed the scope of PeerJ and feel your manuscript does fit within the scope. The manuscript could benefit from some effort to make it more accessible to general readers, and perhaps a supplemental table of specialist terms, but I do not feel you need to drastically remake the paper on this front.
The other comments of reviewer 1 and reviewer 2 seem fair and appropriate to me, so please pay attention to these. In conclusion, I had a hard time distinguishing between 'minor' and 'major' revision, and feel with a bit of work this manuscript can be greatly improved.

Reviewer 1 ·

Basic reporting

The text and figures seem to meet the appropriate standard for PeerJ. No special comments.

Experimental design

The submission describes original primary research. No critical comments for experimental methods.

Validity of the findings

The data and conclusions seem to be appropriately stated. No critical comments.

Additional comments

This paper is as an important contribution to the molecular phylogeny of the south eastern Pacific porcellanids including the type species of the genus Petrolisthes and all known valid species of the related two genera Allopetrolisthes and Liopetrolisthes. I agree with the conclusion that the species of Allopetrolisthes, and Liopetrolisthes are tentatively transferred to the genus Petrolisthes. The south eastern Pacific species of Petrolisthes are probably treated as the “core” members of the genus sensu stricto.

I have made some changes and comments directly on the file of the manuscript using Word's track change feature, for the author to consider (see attached file). The present author should thoroughly check to make sure of the corrections. The comments on the text file are shown by highlighted letters and in parentheses “[ ]”.

A special comment is shown below.

It seems better to take the results of a series of studies on larval morphology into consideration in Discussion. Please look at the following works on the larvae of Petrolisthes and Allopetrolisthes which may additionally support particular clade(s) or sisier speices relationship(s) of your phylogenetic tree or not.

Fujita Y, Shokita S, Osawa M. 2002. Complete Larval Development of Petrolisthes unilobatus reared under laboratory conditions (Decapoda: Anomura: Porcellanidae). Journal of Crustacean Biology 22: 567–580.

Hernández G. 1999. Morfología larvaria de cangrejos anomuros de la Familia Porcellanidae Haworth, 1825 (Crustacea: Decapoda), con una clave para las zoeas de los géneros del Atlántico occidental. Ciencia 7(3): 244–257.

Hernández G, Magán I. 2012. Redescripción de los primeros estadios postembrionarios del cangrejo anomuro Petrolisthes magdalenensis Werding, 1978 (Crustacea: Decapoda: Porcellanidae). Boletín del Instituto Oceanografico de Venezuela 51: 35–51.

Osawa M. 1995. Larval development of four Petrolisthes species (Decapoda, Anomura, Porcellanidae) under laboratory conditions, with comments on the larvae of the genus. Crustacean Research 24: 157–187.

Annotated reviews are not available for download in order to protect the identity of reviewers who chose to remain anonymous.

Reviewer 2 ·

Basic reporting

No comments, see comments to author

Experimental design

No comments, see comments to author

Validity of the findings

No comments, see comments to author

Additional comments

Review: Molecular phylogeny of porcelain crabs (Porcellanidae: Petrolisthes) from the south eastern Pacific: the genera Allopetrolisthes and Liopetrolisthes are not natural entities

General comments;
This study examines the phylogey of porcelaian crabs from the south east Pacific, with two short molecular markers (mt16S and a histone region) for a total of ~600 characters. The central finding is that the genera Allopetrolisthes and Liopetrolisthes are nested well within Petrolisthes, therefore the authors propose re-assigning these sampled species in the genus Petrolisthes. The markers are each very short and the taxonomic sampling is very sparse at the species level, therefore there is no information about species-level variability. The short markers are disconcerting, particularly since the authors only present a total evidence tree, and do not provide trees for each marker. Each tree needs to be presented independently, as well as the total evidence tree; it is not sufficient to say the datasets are not heterogeneous (by what criteria/statistical test?).
It is not made clear why other additional nearly ubiquitous markers such as CO1 were not included in the study; this should be explained. Table 3 is mentioned in the text but was not included in my reviewing materials.
The manuscript is in my option exceedingly dense with specialist terminology, and readers not familiar with crab taxonomic terms (such as myself) will soon become lost. I suggest either including a diagram, a glossary, or explaining each taxonomic term, or submitting this to a more specialist journal since the readership of and scope of PeerJ are very broad. There are several English mistakes and I suggest future submissions should be double checked by an English speaker. This manuscript could possibly be published in PeerJ after dramatic revisions, but only after a resubmission with a major overhaul to appeal to broader readership. The introduction needs to be more focused on the life history, importance, and background for these crabs and less dense with taxonomic terminology… The discussion makes a few overly casual claims about sympatric speciation and ‘evolutionary colonization’, which are speculative and need to be more carefully discussed. As it stands, I think the manuscript would be a better fit for a more specialist taxonomic journal, in spite of the fact that PeerJ has a very broad scope.


Specific comments;
Introduction

Line 57; agonistic? antagonistic?
line 58: should read ‘symbiosis with sea anemones’
line 59; no need for brackets here and should read ‘associated body parts such as bactrophorian setae in the ‘yeti crab’…’
line 87. ‘inward’
line 82. This paragraph consists of one large sentence… how much of this is a direct quotation? I think this could be improved to be more accessible to the general reader, as with the later paragraphs that are loaded with taxonomic terms unfamiliar to anyone not working on crab taxonomy.

Methods
Why was CO1 not sequenced, given that this marker has the largest taxonomic sampling available for the sake of comparison?
Line 176, where is table 3?
Line 197, the reader should be able to see the results of this ‘preliminary analysis’. Why is this not included in the paper? How was heterogeneity accessed? Are these criteria subjective?



Results
Line 249 should read ‘form’

Discussion
Line 285; This line is very speculative and sympatric speciation is very challenging to demonstrate or to rule out competing hypotheses.

Line 319; why does this indicate multiple ‘evolutionary colonization’ event? What is a ‘evolutionary colonization’ event?
Line 342; so to be clear, are you proposing eliminating Liopetrolisthes and Allopetrolisthes? what about P.desmarestii?

---

## Round 0.2 · Minor Revisions

The reviewer is now satisfied with the revisions. I have also been over the manuscript in detail, and found some areas where the English would be well served by being revised or altered. Please see the attached file, and edit the English as you see appropriate. I imagine this revision will take only a few days at most.

Reviewer 2 ·

Basic reporting

No comments

Experimental design

No comments

Validity of the findings

No comments

Additional comments

The manuscript has improved greatly following the suggestions from the previous reviews...

I have only minor suggestions methods line 194-199, suggest use of commas in large numbers.

Line 304 discussion: suggest change to "This suggests that the genus Liopetrolisthes likely diversified in the south eastern Pacific
in sympatry although it remains to be addressed whether or not speciation in the genus Leiopetrolisthes was host driven".

---

## Round 0.3 · accepted · Accept

The manuscript is now acceptable for publication - thank you for your speedy response. I attach a MS Word file here with a few small corrections (font size corrections mainly); please check these during the proof stage.